# Caregiver mental health and school-aged children's academic and socioemotional outcomes: Examining associations and mediators in Northern Ghana

**Marilyn N. Ahun**[1,2]*, **Richard Appiah**[3,4], **Elisabetta Aurino**[5], **Sharon Wolf**[6]

**1** Department of Medicine, McGill University, Montréal, Canada, **2** Department of Global Health and Population, Harvard T.H. Chan School of Public Health, Boston, Massachusetts, United States of America, **3** Department of Psychology, Northumbria University, Newcastle-upon-Tyne, United Kingdom, **4** College of Health Sciences, University of Ghana, Accra, Ghana, **5** School of Economics, University of Barcelona, Barcelona, Spain, **6** Graduate School of Education, University of Pennsylvania, Philadelphia, Pennsylvania, United States of America

* marilyn.ahun@mcgill.ca

**Data Availability Statement:** Data are available upon reasonable request from interested, qualified researchers by contacting the data access

## Abstract

While there is a strong link between caregiver mental health, caregiver engagement, and child development, limited research has examined the underlying mechanisms of these associations in Africa. We examined the mediating role of dimensions of caregiver engagement in the association of caregiver psychological distress with children's academic and socioemotional outcomes in Ghana. Data came from 4,714 children (aged 5–17 years) and their caregivers in five regions of northern Ghana. Caregiver psychological distress and engagement (i.e., engagement in education, emotional supportiveness, and parenting self-efficacy) were self-reported by children's primary caregiver. Children's academic (literacy and numeracy) and socioemotional (prosocial skills and socioemotional difficulties) outcomes were directly assessed using validated measures. Structural equation modelling was used to estimate mediation models. We tested moderation by caregiver exposure to formal education, child's age, and child's sex. Fourteen percent of caregivers experienced elevated psychological distress. Higher levels of psychological distress were associated with children's poorer literacy and numeracy skills, and higher socioemotional difficulties, but not prosocial skills. The mediating role of caregiver engagement varied by caregiver exposure to formal education but not child's age or sex. Caregiver engagement in education explained the association between psychological distress and children's literacy skills (but not numeracy or socioemotional) in families where the caregiver had no formal education (indirect effect: $\beta$ = 0.007 [95% CI: 0.000, 0.016]), explaining 23% of the association. No mediator explained the association of psychological distress with child outcomes among families where the caregiver had some formal education. The mechanisms through which caregiver psychological distress is associated with child outcomes in rural Ghana differ as a function of caregivers' exposure to formal education. These results highlight the importance of developing multi-component and culturally-sensitive programs to improve child outcomes.

committee at IPA Ghana: samadu@poverty-action.org.

**Funding:** This study was supported by the Canadian Institutes of Health Research (Postdoctoral Fellowship #181899 to MNA) and European Research Council (Starting Grant #101041741 to EA). Views and opinions expressed are however those of the authors only and do not necessarily reflect those of the European Union or the European Research Council Executive Agency. Neither the European Union nor the granting authority can be held responsible for them. The intervention the study is based on was funded by the World Bank Strategic Impact Evaluation Fund (#7197944 to EA and SW), Jacobs Foundation (#581254 to EA and SW), Imperial College Research Fellowship (EA), and the EdTech Hub (#0083 to EA and SW). The funders had no role in study design, data analysis, data interpretation, manuscript preparation, or the decision to submit for publication.

**Competing interests:** The authors have declared that no competing interests exist.

Further research in similar contexts is needed to advance scientific understanding on how to effectively promote child and family wellbeing.

## Introduction

Children's healthy development–including cognitive, language, and socioemotional skills–constitutes the building blocks of a healthy, well-functioning, and productive society [1]. In low- and middle-income countries (LMICs), approximately 250 million children are at an increased risk of poor development [2]. Despite this significant burden in LMICs, children living in these contexts remain severely underrepresented in child development and mental health research [3, 4]. Ghana, a lower-middle-income country in West Africa, has a human capital index of 0.44, meaning that children born today are only expected to reach 44% of their developmental potential [5]. Although Ghana has near universal school enrolment rates, education quality and learning outcomes are low [6]. For example, 70% (Grade 2) and 80% (Grade 4) of schoolchildren are unable to read a simple word or perform basic arithmetic operations [5]. Ghanaian children are also at risk of significant mental health problems, with studies reporting prevalence rates of depression (1–18%), anxiety (1–4%), and conduct problems (13%) [7–9] that are relatively higher than some recent global estimates: depression (1.8–2.6%), anxiety (3.4–6.5%), conduct/disruptive disorders (1.5–5.7%) [10, 11].

Exposure to poverty and food insecurity are known to significantly impact children's development and mental health outcomes [1, 2]. In addition to addressing these factors, low-cost solutions targeting modifiable risk and protective factors are urgently needed to improve children's developmental potential and subsequent health and educational outcomes [12]. Two robust and modifiable determinants of children's development are caregiver/parental (henceforth referred to as caregiver for brevity) engagement in learning activities and caregiver mental health [13–15]. Caregiver engagement consists of various processes including a caregiver's belief in their ability to successfully parent their child (i.e., parenting self-efficacy), as well as their provision of emotionally supportive, sensitive, and responsive parenting practices and support for learning. Caregiver engagement has been repeatedly shown to be associated with significant improvements in children's academic and socioemotional outcomes [13–17]. Yet caregiver engagement is low in various settings across Africa (including Ghana), particularly in rural areas [18, 19]. For example, population-level data show that only 14.6% of female and 3.9% of male caregivers in sub-Saharan Africa engaged in adequate levels of caregiver engagement (defined as 4 out of 6 play and stimulation activities) with their 2-to-4-year-old child in the past 3 days compared to 39.8% of female and 11.9% of male caregivers across 62 other LMICs [18]. The most recent data from Ghana indicate even lower rates of female (11.2%) and male (3.1%) caregiver engagement [20].

One potential explanation for the low level of caregiver engagement in various contexts of Africa is caregiver mental health burden. The experience of mental health problems can negatively influence caregivers' beliefs in their parenting abilities, as well as their capacity to engage in emotionally supportive and responsive interactions with their child. Caregiver mental health problems and their association with less positive parenting practices can subsequently increase the risk of worse child outcomes [21, 22]. Given that the prevalence of mental health problems (e.g., depression) in female caregivers is significantly higher in LMICs (19–25%) compared to high-income countries (HICs; 11–13%) [23, 24], interventions that aim to improve caregiver engagement in these contexts need to consider the extent to which

caregivers' mental health may influence engagement and thus, its potential impacts on child outcomes. Mental health problems are also prevalent in male caregivers (7–13%), however, most of these data are from HICs [25].

It is also important to consider caregivers' mental health given strong evidence of its consistent negative associations with children's academic and socioemotional outcomes in HIC and LMIC contexts [23, 26]. For example, data from various countries, including Brazil, Canada, South Africa, and the UK, consistently show that children of mothers who experience depression and anxiety are more likely to experience mental health problems such as depression, anxiety, and suicidal ideation and attempt in childhood and adolescence [27–32]. Maternal depression and anxiety are also consistently associated with poorer child and adolescent cognitive/academic outcomes across various contexts [26]. Fewer studies have examined the association of paternal mental health with child and adolescent outcomes, but a recent umbrella review–consisting primarily of studies in HICs–suggests that children of fathers who experience depression and anxiety are more likely to experience mental health problems in childhood and adolescence [33]. Even fewer studies have focused on academic outcomes, with two studies (Norway and USA) reporting language difficulties in children whose fathers experience depression [34, 35].

Emerging research from HICs suggests that different dimensions of caregiver engagement mediate the association between caregiver–primarily maternal–mental health and children's developmental outcomes [21]. Yet, there is very little research from LMICs, particularly in West African contexts [36]. The handful of studies conducted in LMICs (i.e., Bangladesh, Ghana, Liberia, South Africa, Vietnam) have identified discipline, positive parent-child interactions, parenting self-efficacy, maternal intrusiveness and coerciveness, and maternal responsiveness as significant mediators of the associations of caregiver mental health with children's cognitive/academic and socioemotional outcomes [37–41]. Further research is needed to better understand why and for whom caregiver mental health problems increase the risk of poorer child outcomes.

An understanding of the mechanisms of this association would help identify and address factors that promote healthy caregiver-child relationships and inform appropriate support programs to ensure better child and family outcomes in LMIC contexts [12, 36]. This study aims to address gaps in the literature by using longitudinal data to examine the mediating role of different dimensions of caregiver engagement in the association of caregiver mental health with children's academic and socioemotional outcomes in Ghana.

## Methods

### Study population

Participants comprised primary caregivers in northern Ghana (Northern, North East, Savannah, Upper East, and Upper West regions) who participated in an intervention study in 2021. The study was a household-randomized trial of a messaging program aimed at supporting caregiver educational engagement and child schooling in the wake of the COVID-19 pandemic through SMS-text nudges to participants residing in rural, low-literacy community settings [42]. This context was chosen because known barriers to caregiver engagement (i.e., cognitive, behavioural, and information barriers that amplify resource or time limitations) can be very high in low-resource contexts such as the one of northern Ghana, where half of the population lives below the poverty line [43]. Additionally, many children in these communities are first-generation learners, and there are significant gender gaps in education [44]. Households were sampled from two previously completed studies and were eligible to be included in the intervention if there was an adult and at least one school-aged child (aged 5–17 years) in the

household, and if they consented to participate in the intervention. Two focal children in each household, one younger child (5–9 years) and one older child (10–17 years) were randomly sampled. The analytic sample consisted of 4,714 children (2,357 families) with available data on the variables of interest.

## Study site and design

Details of the intervention's design, methodology, and primary impacts have been published [42]. Primary caregivers living in five regions of northern Ghana were recruited through a phone-based baseline survey from December 1st to 31st 2020. After consenting to participate, families were randomly assigned to one of the treatment conditions or the control group. The treatment and control groups were statistically equivalent across a broad range of baseline characteristics. The only significant differences were that a greater number of control families owned a television, and treatment families were more likely to live in the North East region and had low educational aspirations for male children [42].

## Intervention

Caregivers in the intervention arms received biweekly SMS messages beginning in January 2021 and ending 12 or 24 weeks from onset, depending on randomized assignment to the 'short' or 'long' intervention groups. The messages aimed to empower caregivers through information, reminders, and suggestions of practical, non-academic activities to engage with their children's education. Messages also targeted fostering social-emotional skills, with themes such as positive discipline, growth mindset, and warm communication at home. Data on child and caregiver outcomes were collected in midline and endline in-person surveys at participant homes in April-June and August-September 2021, respectively. The independent, dependent, and mediator variables used in the present study were assessed at endline.

The intervention led to improved caregiver engagement and children's socioemotional skills and school attendance, but only in families where caregivers had some formal education [42]. In families where caregivers had no formal education, the intervention had small negative effects on both caregiver engagement and child outcomes.

## Ethics statement

The study's protocol was reviewed and approved by the Institutional Review Board at Innovation for Poverty Action 2020, a global non-governmental organization with an office in Ghana (#15573). Informed written consent was obtained from all adult participants and from the parent/legal guardian of all participants below 18 years of age.

## Caregiver psychological distress

Caregivers self-reported their mental health using the Kessler Psychological Distress Scale [45]. This scale is a 10-item questionnaire used globally to measure general psychological distress based on questions about how frequently an individual experienced anxiety and depressive symptoms in the past four weeks, and has previously been used and validated in Ghana [46, 47]. Each item is scored from zero (none of the time) to four (all of the time). Items are summed to create a total score, where higher scores indicate higher psychological distress and an increased likelihood of a mental health disorder ($\alpha = 0.86$). Efforts to identify an optimal cut-off score that indicates clinically significant psychological distress have produced mixed results across HICs and LMICs [48]. We therefore define elevated psychological distress in this sample as a score greater than or equal to one standard deviation above the mean.

## Children's academic and socioemotional skills

To accommodate the wide age range of children in the sample, three different assessments that varied the number and difficulty of items based on child age were used to measure children's academic skills. Literacy skills (i.e., expressive language, non-word reading, spelling, oral reading and comprehension, and phonological awareness) were measured using age-specific subtasks of the International Development and Early Learning Assessment (IDELA) [49], the Early Grade Reading Assessment [50], and items from the Young Lives surveys [51]. Numeracy skills (i.e., number identification, number/quantity discrimination, missing number patterns, number sorting, word problems, and operations such as addition/subtraction, multiplication, and division) were also assessed with the IDELA and items from the Young Lives survey, as well as the Early Grade Math Assessment [52]. IDELA is a global tool designed for use across different contexts and has been validated in various LMICs and other African countries, including Ghana [53–55]. To create a summary score for literacy ($\alpha = 0.65$) and numeracy ($\alpha = 0.70$) skills, the total correct number of answers on each subtask was summed within three age groups (5–9, 10–14, and 15–17 years) and standardized by age (mean = 0, SD = 1) to represent children's performance relative to peers in their age group.

Children's socioemotional skills were assessed with a caregiver-report and a direct assessment measure. Caregivers reported the extent to which each child exhibited socioemotional difficulties in the last six months using the Strengths and Difficulties Questionnaire ($\alpha = 0.71$), which has been validated in the Ghanaian context [56, 57]. In addition, children's prosocial conflict resolution skills and relationships (i.e., how much children draw on relational support from caregivers and other community members in challenging times) ($\alpha = 0.40$) were assessed using the International Social-Emotional Learning Assessment, a tool specifically developed for use in low-resource contexts and previously validated in other African countries [58, 59].

## Mediators: Caregiver engagement in education, emotional supportiveness, and parenting self-efficacy

Caregiver engagement in education ($\alpha = 0.74$) was operationalized as their self-reported engagement in a set of six activities related to their child's learning and education over the past three days. Activities were slightly different for older and younger children to ensure developmental appropriateness. For younger children (ages 5–9), the home stimulation measure from UNICEF's Multiple Indicators Cluster Survey (MICS) was used: reading or looking at books, telling stories, singing songs, taking the child outside the home, playing with the child, and naming/counting/drawing with the child. For older children (ages 10–17), activities were adapted from the UNICEF MICS and Young Lives surveys [51]: working on a project together, playing sports/active games/exercise, discussing time management, talking about family/community history/heritage, discussing future education and career plans, and encouraging the child to listen to or watch remote teaching. Both the MICS and Young Lives Survey have been validated in other African countries, including Ghana (MICS) [20, 51].

Caregivers reported the extent of emotional support they provided their children using a 5-item scale from the Early Childhood Longitudinal Study-Kindergarten Cohort [60]. They answered a variety of hypothetical questions specific to supporting their children's emotional needs on a 4-point scale (1 = very often true, 2 = often true, 3 = sometimes true, and 4 = never true). Example items include: "Even if I am really busy, I make time to listen to [child]," and "I encourage [child] to talk about his/her troubles" ($\alpha = 0.69$). Items were adapted to the specific child age groups in the intervention (i.e., 5–9 and 10–17 years).

Finally, caregivers' parenting self-efficacy was self-reported using the 8-item subscale related to self-efficacy regarding their children's schooling and learning from Bandura's

Parental Self-Efficacy Scale [61]. The scale was scored from 1–5, with 1 = nothing, and 5 = a great deal. Example items include: "How much can you do to make your children see school as valuable?" and "How much can you do to help your children get good grades in school?" ($\alpha = 0.89$).

## Data analysis

Simple, unadjusted, linear regressions were used to test associations between the independent, mediator, and dependent variables. We then used structural equation modelling (SEM) to test the potential mediating role of caregiver engagement in education, emotional supportiveness, and parenting self-efficacy in the associations of caregiver psychological distress with children's literacy, numeracy, and prosocial skills, as well as their socioemotional difficulties. A separate model was estimated for each child outcome. The mediation models controlled for intervention status (control vs treatment arms), and midline levels of the mediators and dependent variables. We also controlled for baseline levels of caregiver engagement in education; none of the other variables were assessed at baseline. Mediation was tested via the significance of the indirect effect from the independent variable via the mediators to each outcome [62]. The indirect effect was considered significant if the product of the coefficient of the pathway from the independent variable to the mediator and the coefficient of the pathway from the mediator to the dependent variable was significant [63]. Two-sided p<0.05 indicated statistical significance. The term 'effect' is used in line with SEM nomenclature to describe associations between variables [64]. Given the cross-sectional nature of these data, we acknowledge that these 'effects' are not causal. The Huber-White sandwich estimator [65] was used to account for the non-independence of observations by adjusting standard errors for clustering of data at the family level [66].

Given the divergent impacts of the intervention on child outcomes as a function of caregivers' exposure to formal education, we tested whether this moderated the tested associations by comparing the fit of a freely estimated model (i.e. all estimated parameters allowed to freely vary across families where the primary caregiver had some formal education and families where the caregiver had no formal education) with the fit of a model in which all estimated parameters were constrained to be equal across families with different levels of caregiver formal education [64]. As prior empirical evidence–from Ghana and elsewhere–indicates that there may be child age and sex differences in the associations between caregiver psychological distress, the different dimensions of caregiver engagement, and children's academic and socioemotional skills, we also tested child age and child sex as potential moderators [17, 21, 67, 68].

SEM was performed with Mplus version 8.6 [69] using the robust maximum likelihood estimator (MLR). Because bootstrapped confidence intervals of the indirect effect are not available in Mplus for this analysis, the R-Mediation package was used to build unbiased confidence intervals for indirect effects [70]. Furthermore, differences in fit between the freely estimated and constrained models were calculated by hand using the equation specified by Satorra and Bentler, since the likelihood ratio test cannot be used for models estimated with MLR [71]. Adequate model fit for the final models was considered using a root mean square error of approximation (RMSEA) of <0.06 and confirmatory fit index (CFI) of > 0.90 [72, 73]. Missing data were handled using the full information maximum likelihood. Descriptive statistics and unadjusted regressions were obtained using the Statistical Packages for the Social Sciences version 25 [74]. The MplusAutomation package was used in R version 4.0.3 to prepare the data for use in Mplus [75, 76].

## Results

On average, caregivers were 42 years old, and households typically consisted of 11 members, including 3 school-age children. Forty-seven percent of sampled caregivers were male, and only 28% had some formal education. A binary variable indicating that caregivers had some (=

**Table 1. Sample characteristics by caregiver's exposure to formal education.**

| | Entire sample (n = 4,714) | Caregiver formal education | |
| --- | --- | --- | --- |
| | | No formal education (n = 3,403) | Some formal education (n = 1,311) |
| Child age, *mean (SD)* | | | |
| 5-9-year-olds | 7.34 (1.3) | 7.36 (1.2) | 7.27 (1.3) |
| 10-17-year-olds | 12.95 (2.2) | 12.93 (2.2) | 12.98 (2.3) |
| Child sex, % | | | |
| Female | 41.4 | 41.0 | 41.8 |
| Male | 58.6 | 59.0 | 58.2 |
| Primary caregiver, % | | | |
| Female caregiver | 52.6 | 54.1* | 48.7* |
| Male caregiver | 47.4 | 45.9* | 51.3* |
| Caregiver age, *mean (SD)* | 42.44 (11.1) | 43.22 (10.7)* | 40.41 (11.8)* |
| Caregiver marital status, % | | | |
| Married | 94.9 | 95.4 | 93.5 |
| Other (e.g., single, divorced) | 5.1 | 4.6 | 6.5 |
| Caregiver number of children, *mean (SD)* | 3.24 (1.8) | 3.22 (1.7) | 3.30 (1.9) |
| Caregiver ethnicity, % | | | |
| Dagomba | 97.4 | 97.4 | 97.3 |
| Other (e.g., Frafra, Maprusi, Gonja) | 2.6 | 2.6 | 2.7 |
| Caregiver level of formal education, % | | | |
| Pre-primary | - | | 5.4 |
| Primary | | | 42.3 |
| Junior High School | | - | 25.7 |
| Senior High School | | | 17.1 |
| Post-secondary | | | 9.1 |
| Household size, *mean (SD)* | 11.10 (5.8) | 11.15 (1.7) | 10.98 (6.0) |
| Household intervention status, % | | | |
| Intervention arms | 78.4 | 78.1 | 79.4 |
| Control arm | 21.6 | 21,9 | 20.6 |

*Differences between caregivers with no formal education versus caregivers with some formal education is significant at *p<0.05* according to chi-square (for categorical variables) and ANOVA (for continuous variables) tests

1) versus no (= 0) formal education was used to distinguish families. Sampled children were, on average, 10 years old, and were almost evenly distributed into two target age groups: 45% 5-9-year-olds and 55% 10-17-year-olds. Most families are of Dagomba ethnicity (93%), which is part of the second most dominant ethnic group (Mole-Dagbani) in Ghana [77]. The primary caregiver in families where the caregiver had some formal education was more likely to be male (51 vs 46%) and younger (40 vs 43 years) compared to families where the caregiver had no formal education. Descriptive statistics for the whole sample and by caregiver exposure to formal education are presented in Table 1.

There was no significant difference ($\chi^2$ = 1.99, p = 0.157) in the prevalence of elevated caregiver psychological distress among caregivers who had no formal education (15%) and those who had some formal education (13%). In unadjusted regression models, caregiver psychological distress was significantly negatively associated with parenting self-efficacy ($\beta$ = -0.08, $p<0.05$), as well as children's literacy ($\beta$ = -0.06, $p<0.05$) and numeracy ($\beta$ = -0.07, $p<0.05$) skills. It was also significantly positively associated with children's socioemotional difficulties ($\beta$ = 0.12, $p<0.05$). However, caregiver psychological distress was not significantly associated

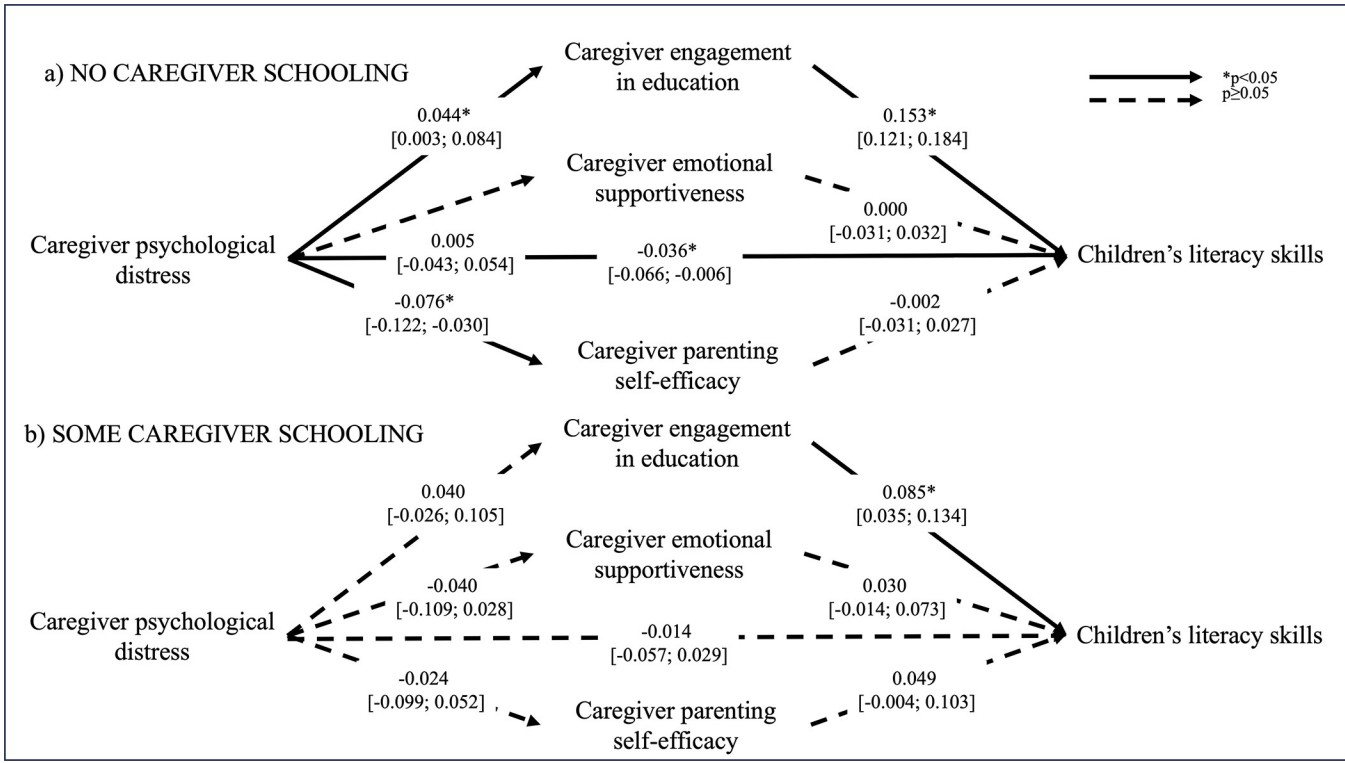

**Fig 1. Mediation models of the adjusted associations (standardized β [95% confidence intervals]) between caregiver psychological distress and children's literacy skills across different levels of caregiver formal education.** Models control for intervention status (control vs treatment arms), baseline caregiver engagement in education, and midline caregiver engagement in education, emotional supportiveness, and parenting self-efficacy, and children's literacy skills.

with the remaining variables (caregiver emotional supportiveness [$β$ = -0.01, $p = 0.394$] and child prosocial skills [$β$ = 0.01, $p = 0.374$]), although it was marginally statistically significantly and positively associated with caregiver engagement in education ($β$ = 0.03, $p = 0.063$). Higher levels of caregiver engagement in education, emotional supportiveness, and parenting self-efficacy were significantly associated with children's improved literacy, numeracy, and prosocial skills (data not shown). Caregiver engagement in education ($β$ = -0.06, $p<0.05$) and parenting self-efficacy ($β$ = -0.03, $p<0.05$)–but not emotional supportiveness ($β$ = 0.00, $p = 0.993$)–were also significantly associated with higher levels of children's socioemotional difficulties.

Tests of differences in associations between families where the primary caregiver had some formal education and families where the caregiver had no formal education showed that mediation models where parameters were freely estimated were superior to models with constrained parameters: literacy skills ($\chi^2_{57}$ = 3432.9, $p<0.05$), numeracy skills ($\chi^2_{57}$ = 3485.6, $p<0.05$), prosocial skills ($\chi^2_{57}$ = 3519.8, $p<0.05$), socioemotional difficulties (The degrees of freedom are different for the model with socioemotional difficulties because this variable was only assessed at endline, and we therefore could not control for midline assessment in the model. All other models included a control variable of the child outcome assessed at midline) ($\chi^2_{47}$ = 3162.8, $p<0.05$). The final models were therefore estimated separately for families with different levels of caregiver formal education. Figs 1–4 illustrate the mediation models across caregiver exposure to formal education. The total and simple direct and indirect effects of each model are reported in Table 2.

In families where caregivers had no formal education, children exposed to higher levels of caregiver psychological distress had poorer literacy skills (RMSEA = 0.034, CFI = 0.979), and

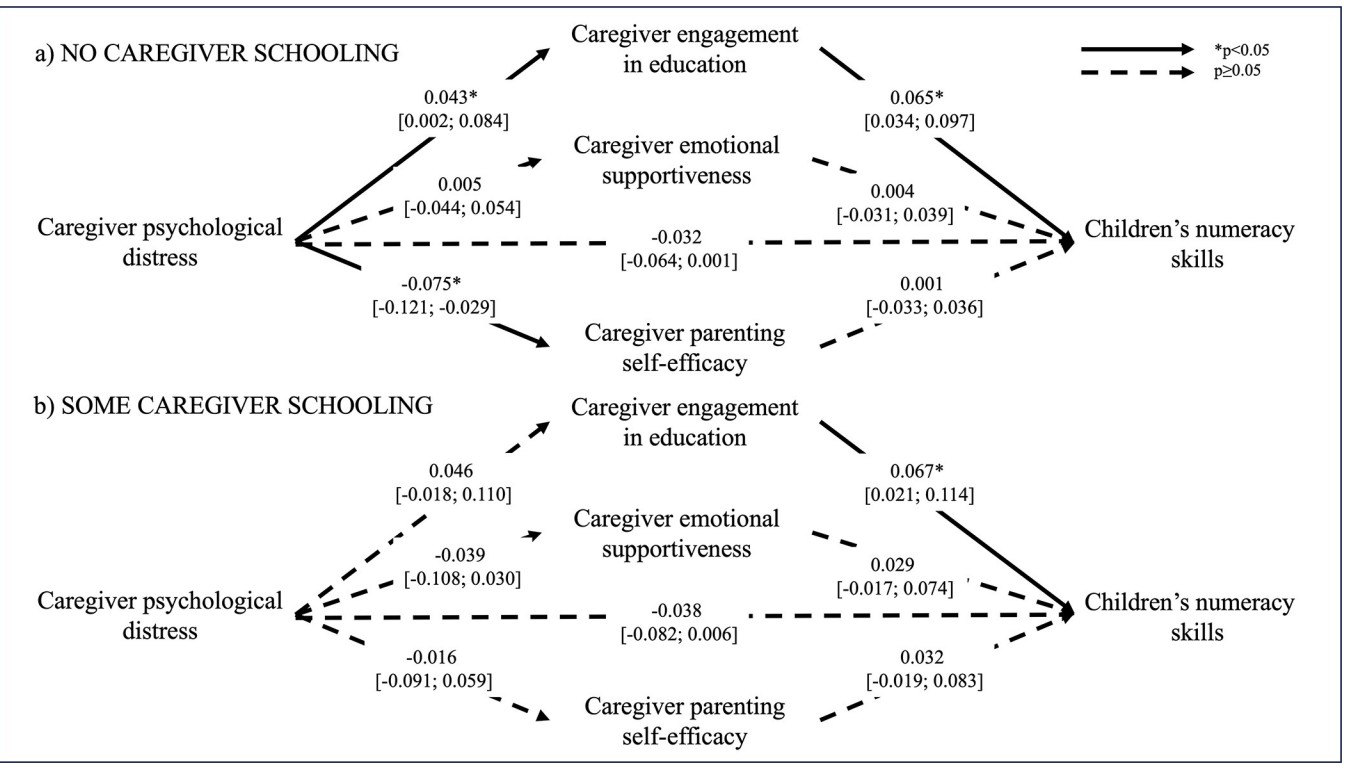

**Fig 2. Mediation models of the adjusted associations (standardized β [95% confidence intervals]) between caregiver psychological distress and children's numeracy skills across different levels of caregiver formal education.** Models control for intervention status (control vs treatment arms), baseline caregiver engagement in education, and midline caregiver engagement in education, emotional supportiveness, and parenting self-efficacy, and children's numeracy skills.

this association was partially mediated by caregiver engagement in education (*p<0.05*). Specifically, caregiver engagement in education was positively associated with caregiver psychological distress and explained 23% of the association between caregiver psychological distress and children's literacy. Caregiver psychological distress was associated with higher levels of socioemotional difficulties in children (RMSEA = 0.032, CFI = 0.920), but none of the mediators explained this association. Finally, although caregiver engagement in education was significantly associated with both caregiver psychological distress and children's numeracy skills (RMSEA = 0.035, CFI = 0.977), the indirect effect did not reach significance. None of the other hypothesized associations in the mediation models were significant. Caregiver psychological distress was not associated with children's prosocial skills (RMSEA = 0.029, CFI = 0.955).

With respect to families where caregivers had some formal education, children exposed to higher levels of psychological distress also experienced more socioemotional difficulties, but this association was not explained by any of the proposed mediators. Caregiver psychological distress was not significantly associated with any of the other child outcomes and there was thus no evidence of mediation. Neither child age nor child sex were significant moderators of the mediation models for any child outcome (data not shown).

## Discussion

This study addresses multiple gaps in child development and mental health research by examining the associations between caregiver psychological distress, caregiving processes, and children's academic and socioemotional outcomes in Ghana. Firstly, we used robust statistical

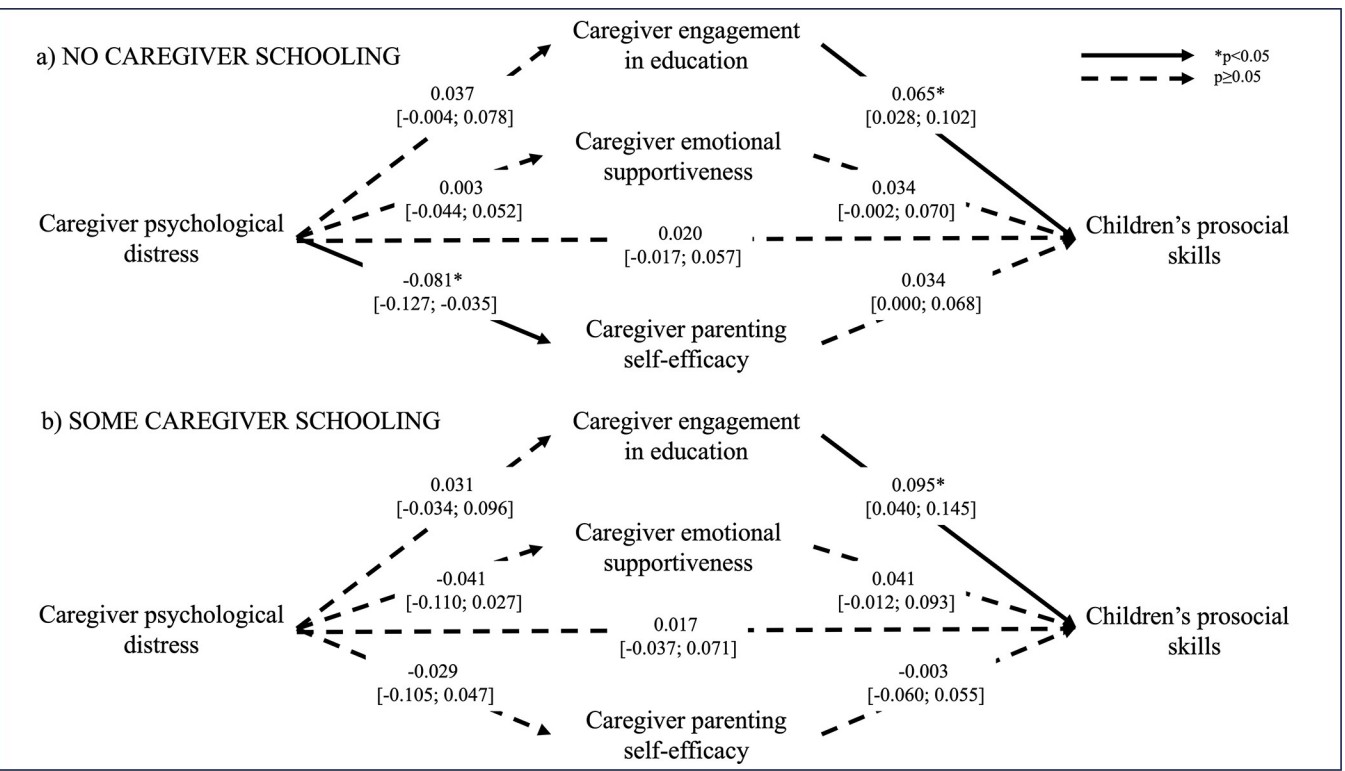

**Fig 3. Mediation models of the adjusted associations (standardized β [95% confidence intervals]) between caregiver psychological distress and children's prosocial skills across different levels of caregiver formal education.** Models control for intervention status (control vs treatment arms), baseline caregiver engagement in education, and midline caregiver engagement in education, emotional supportiveness, and parenting self-efficacy, and children's prosocial skills.

modelling to test existing hypotheses of the potential mechanisms through which caregiver psychological distress may be associated with child outcomes by examining different dimensions of caregiver engagement as mediators. We found that caregivers' psychological distress was significantly associated with worse literacy and numeracy skills and increased socioemotional difficulties in children, and that caregivers' active engagement in their children's learning and education explained almost a quarter of the association with children's literacy skills. Secondly, by using data from an intervention promoting caregiver engagement in children's learning, we shed light on the complex interactions between caregiver characteristics and their subsequent associations with child outcomes. Specifically, we tested the moderating role of caregiver exposure to formal education and found that the reported mediation in the association with children's literacy skills was only significant among families where caregivers had no formal education. Finally, by focusing on families in rural areas in a lower-middle-income country, this study contributes to diversifying research on child development and mental health and showing that caregiver engagement in education–but not emotional supportiveness or parenting self-efficacy–matters in a context with low levels of formal education.

Our results from a sample of male and female caregivers replicate and extend findings documenting the association of female caregivers' mental health with child and adolescent academic and socioemotional outcomes across various contexts. Additionally, this is one of the few studies to demonstrate that male caregivers' mental health is significantly associated with poorer academic and socioemotional outcomes in an LMIC context. These findings suggest that both maternal and paternal mental health play a role in children's development and

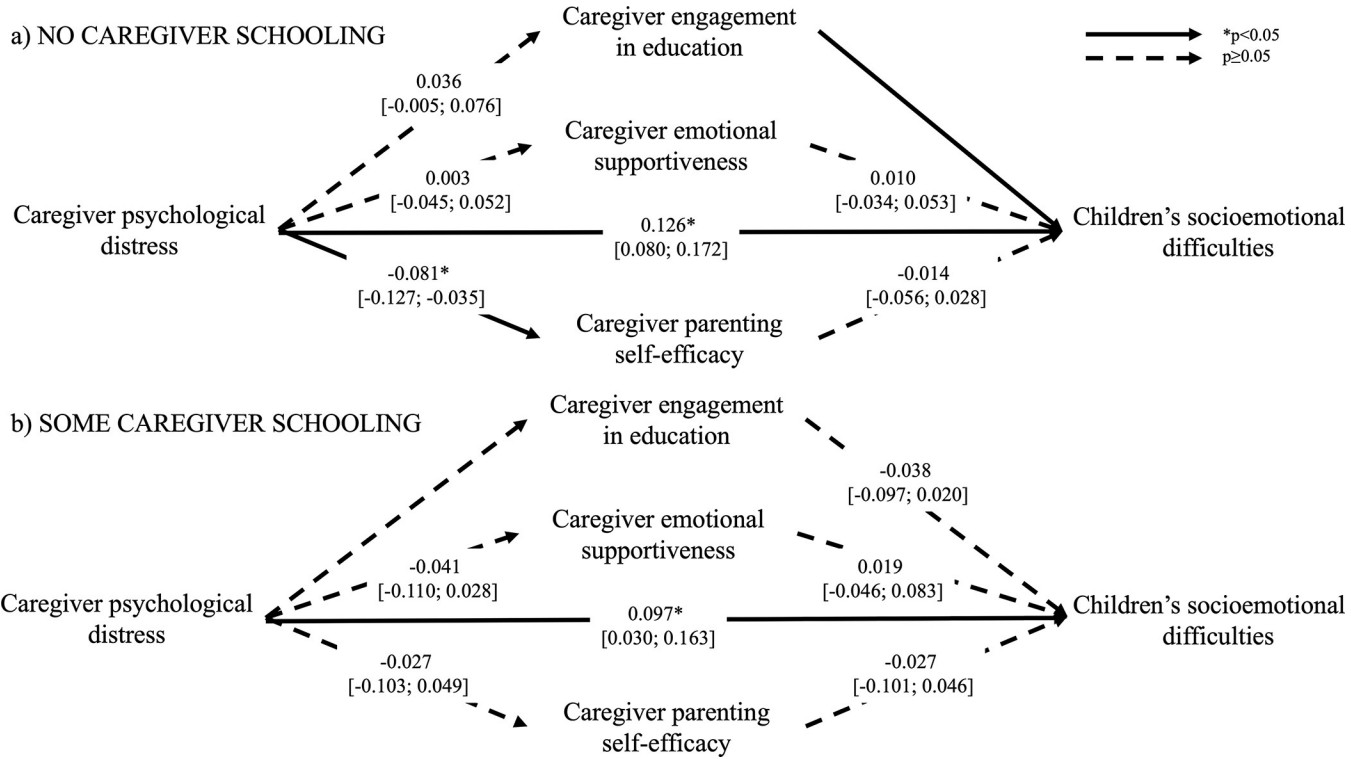

**Fig 4. Mediation models of the adjusted associations (standardized β [95% confidence intervals]) between caregiver psychological distress and children's socioemotional difficulties across different levels of caregiver formal education.** Models control for intervention status (control vs treatment arms), baseline caregiver engagement in education, and midline caregiver engagement in education, emotional supportiveness, and parenting self-efficacy.

underscore the importance of including both female and male caregivers in interventions to support family wellbeing and child development. Our findings also suggest that targeting caregiver engagement in education in families where caregivers have no formal education could be a fruitful way to improve academic outcomes for children and adolescents.

The handful of other studies that have examined the mechanisms of associations between caregiver mental health and child outcomes across West Africa have identified other dimensions of caregiver engagement that could be targeted in interventions. Using data from an urban Ghanaian sample of male and female caregivers of school-aged children (3–10 years), Huang and colleagues found that mild physical discipline and psychological discipline mediated the association between depression in caregivers and children's externalizing symptoms and attention problems [38]. Another study, focusing on a low-income urban sample in Liberia, identified positive caregiver-child interactions (e.g., physical affection, praise) as a mediator of the association between male and female caregivers' depression and children's emotional and behavioural wellbeing [39]. The study in Ghana also tested caregiver engagement in children's education as a potential mediator but found that it was only associated with child outcomes, not parental depression [38]. This could be due to the relatively high education level of parents (66% had completed secondary school or higher), as we also found no significant associations between caregiver psychological distress and caregiver engagement in education in the sub-sample of parents with some formal education.

These findings could be explained by the *leveraging* or *accumulated advantages* hypotheses, whereby caregivers with relatively higher levels of formal education (a proxy for higher socioeconomic status) have more resources to buffer the negative associations between their mental

**Table 2. Indirect and direct effects[a] (standardized β [95% confidence intervals]) of caregiver psychological distress on children's academic and socioemotional skills[b].**

| | No formal education | Some formal education |
|---|---|---|
| *LITERACY* | | |
| Total effect | -0.029 [-0.059, 0.000] | -0.013 [-0.057, 0.031] |
| Direct effect | -0.036 [-0.066, -0.006]* | -0.014 [-0.057, 0.029] |
| Indirect effect | | |
| Via caregiver education in engagement | 0.007 [0.000, 0.016]* | 0.000 [-0.002, 0.002] |
| Via caregiver emotional supportiveness | 0.000 [-0.002, 0.002] | -0.001 [-0.005, 0.001] |
| Via caregiver parenting self-efficacy | 0.000 [-0.002, 0.002] | -0.001 [-0.006, 0.003] |
| *NUMERACY* | | |
| Total effect | -0.029 [-0.063, 0.004] | -0.038 [-0.084, 0.008] |
| Direct effect | -0.032 [-0.065, 0.001] | -0.039 [-0.085, 0.006] |
| Indirect effect | | |
| Via caregiver education in engagement | 0.003 [0.000, 0.006] | 0.003 [-0.001, 0.009] |
| Via caregiver emotional supportiveness | 0.000 [-0.001, 0.001] | -0.001 [-0.005, 0.001] |
| Via caregiver parenting self-efficacy | -0.000 [-0.003, 0.003] | -0.001 [-0.004, 0.003] |
| *PROSOCIAL SKILLS* | | |
| Total effect | 0.020 [-0.018, 0.057] | 0.019 [-0.036, 0.073] |
| Direct effect | 0.020 [-0.017, 0.057] | 0.017 [-0.037, 0.071] |
| Indirect effect | | |
| Via caregiver education in engagement | 0.002 [-0.000, 0.006] | 0.003 [-0.003, 0.010] |
| Via caregiver emotional supportiveness | 0.000 [-0.002, 0.002] | -0.002 [-0.007, 0.001] |
| Via caregiver parenting self-efficacy | -0.003 [-0.001, 0.000] | -0.000 [-0.003, 0.003] |
| *SOCIOEMOTIONAL DIFFICULTIES* | | |
| Total effect | 0.121 [0.076, 0.166]* | 0.093 [0.029, 0.158]* |
| Direct effect | 0.122 [0.077, 0.167]* | 0.095 [0.030, 0.160]* |
| Indirect effect | | |
| Via caregiver education in engagement | -0.002 [-0.006, 0.000] | -0.001 [-0.006, 0.002] |
| Via caregiver emotional supportiveness | 0.000 [-0.001, 0.001] | -0.001 [-0.005, 0.003] |
| Via caregiver parenting self-efficacy | 0.001 [-0.002, 0.005] | -0.001 [-0.002, 0.005] |

[a] The term 'effect' is used in line with SEM nomenclature to describe associations between the exposure, mediator, and outcome variables [64]. Given the cross-sectional nature of these data, we acknowledge that these 'effects' are not causal

[b] All models adjusted for intervention status (control vs treatment arms), midline levels of caregiver engagement in education, emotional supportiveness, and parenting self-efficacy, and baseline caregiver engagement in education. The midline value of each child outcome was additionally controlled for in the respective model of each outcome, except for socioemotional difficulties which was not assessed at midline.

*p<0.05

health and their engagement in their children's learning [78, 79]. This suggests that families from lower socioeconomic backgrounds are more at risk of experiencing negative associations between caregiver mental health and caregiver engagement or other parenting practices, likely due to the cumulation of stressful life events. This hypothesis is supported both by our finding of a significant association between these factors in the sub-sample where caregivers had no formal education, and that of the Liberian study, which focused on a low-income sample [39].

The importance of caregivers' level of formal education in understanding the interactions between caregiver and child outcomes was also evident in the intervention from which our

data were taken, whereby the intervention improved caregiver and child outcomes in families where caregivers had some formal education but had negative impacts in families with a primary caregiver who had no formal education [42]. Quantitative examinations of these findings revealed that caregivers with no formal education interpreted intervention messages as a signal that they were not supporting their children well enough, thus leading them to lower their educational aspirations for their children [42]. It is also possible that caregivers' negative interpretation of intervention messages adversely impacted their motivation and sense of capability–two necessary prerequisites for messaging to work [80]–to enact the information conveyed in the messages. It therefore seems that for these caregivers, the intervention may have made their own education and resource limitations more salient, without addressing other barriers, leading to overall disengagement [42].

Additionally, our finding of associations between caregiver psychological distress and caregiver engagement in education–as well as their parenting self-efficacy–in this sub-sample could partly reflect the fact that the intervention did not address caregiver mental health, a known determinant of caregivers' parenting practices and self-efficacy [21, 22, 79]. Overall, these findings point to the importance of understanding the context within which families live and acknowledging the role of multiple intersecting factors that significantly impact children's development and mental health when designing interventions to improve child outcomes [12, 81]. They also underscore the importance of designing, implementing, and evaluating the impact of multi-component interventions to successfully enhance caregiver engagement and improve caregiver mental health to promote child development and family wellbeing [82, 83]. A recent global systematic review of such interventions found that although they improve children's cognitive and socioemotional outcomes, there is no meta-analytic effect on depressive symptoms in female caregivers [84]. This is likely explained by the fact that interventions are overwhelmingly focused on improving caregiver engagement and include little content addressing caregiver mental health. To fully realize the potential of multi-component interventions to support children and their families in various contexts, we need to design programs that adequately address caregiver engagement and caregiver mental health.

We found no mediating role of other dimensions of caregiver engagement–emotional supportiveness and parenting self-efficacy–in the associations between caregiver psychological distress and child outcomes. Although studies have consistently found associations between caregiver mental health–particularly depression–and parenting self-efficacy [22], and between parenting self-efficacy and children's developmental outcomes [16], only a handful of studies have formally tested the mediating role of parenting self-efficacy in the caregiver mental health-child development association, most of them in HICs [16]. These few studies suggest that caregivers' belief in their parenting abilities does explain some portion of the association between their mental health and their children's developmental outcomes [16, 41], but further research is needed to strengthen this evidence base. It is worth noting that these studies focused on very young children (<3 years old) and there is some evidence of a stronger association between caregiver mental health and parenting self-efficacy in the first few years postpartum [22]. It is therefore likely that we found no mediating role of parenting self-efficacy in our sample due to the older age of children (5–17 years old).

With respect to emotional supportiveness, a recent meta-analysis on the mediating role of parenting in the association between maternal depression and child outcomes found that positive parenting practices (including emotional supportiveness) had weaker associations with maternal depression compared to more global measures of parenting [21]. Additionally, the association between positive parenting practices and various child outcomes were strongest for children aged 0–12 months. Our finding that emotional supportiveness did not mediate the association between caregiver psychological distress may therefore be explained by the

older age of children in our sample. However, as most studies in the meta-analysis were based in HICs, further research in LMICs is needed to better understand the role emotional support-iveness–and other dimensions of positive parenting practices–may play in this association, especially in rural and low-literacy contexts.

The results of Goodman and colleagues' [21] meta-analysis of parenting as a mediator of the association between maternal depression and child outcomes also lend support to our remaining findings. As in our study, neither child's sex nor child's age moderated the meta-analytic mediation model [21]. Although there is prior evidence of children's age moderating the association of caregiver engagement in education with children's developmental outcomes in Ghana [17], this sample focused on younger children (3-4-year-olds), and the findings may thus not be applicable to older children. Additionally, Goodman and colleagues [21] did find that the domain of child development moderated the meta-analytic mediation model, whereby different dimensions of parenting did not mediate the association of maternal depression with children's attachment, emotional/behavioural, and cognitive vulnerability outcomes [21]. This mirrors our finding that none of the caregiver engagement dimensions mediated the associa-tion of caregiver psychological distress with children's numeracy and prosocial skills and socioemotional difficulties. Overall, these findings suggest that there are no differences amongst boys and girls or children of different ages in the associations between caregiver psy-chological distress, different dimensions of caregiving, and children's academic and socioemo-tional outcomes. Further research is needed to understand whether this translates into a universal approach for interventions targeting these factors.

This study has many strengths including the use of a large sample from a randomized con-trolled trial, robust analytic method to address key gaps in the literature in a novel context, and both self-reported and observational outcomes, as well as the inclusion of participants living in a rural, low-literacy area–a context underrepresented in research. However, it is not without its limitations. Firstly, although our analyses are based on strong empirical evidence on the nature of associations between caregiver mental health, caregiver engagement, and child out-comes, the use of cross-sectional data precludes us from making any claims about the causality or directionality of the reported findings. Secondly, both caregiver and child mental health outcomes were not clinically assessed, thus limiting our capacity to make inference to popula-tions with clinically significant levels of mental health problems. Thirdly, the measure used for children's prosocial skills had low reliability in our sample, suggesting that it did not consis-tently assess children's prosociality. These results should therefore be interpreted with caution. Finally, the reported associations were generally small, suggesting that other dimensions of caregiving (e.g., responsivity and sensitivity in caregiver-child interactions, disciplinary strate-gies) would be important to consider in future research. It is worth noting, though, that small associations are still relevant for large-scale interventions, as small differences at a population level can greatly impact population health [85].

We also note that our findings should be interpreted in the context of the COVID-19 pan-demic as data were collected during its acute phase. Indeed, it is possible that caregivers–and their children–were experiencing higher levels of overall distress at this time, as well as distress due to lost incomes and school closures (schools in Ghana were closed for 10 months begin-ning in March 2020) [86], all of which could have influenced the nature of reported associa-tions between caregiver engagement and child outcomes. However, it is worth noting that the prevalence of psychological distress in our sample (14%) is comparable to prevalence rates of depression (16.8–19.7%) and anxiety (11.4%) in Ghanaian caregivers in pre-COVID-19 stud-ies [87–89]. Continued research on these associations is clearly needed to better understand their nature and consistency beyond the COVID-19 pandemic.

## Conclusion

This study sheds light on the mechanisms through which caregiver mental health is associated with children's development in Ghana, as well as how mechanisms differ as a function of caregivers' exposure to formal education. Our findings suggest that addressing caregiver mental health could be a viable component of interventions targeting caregiver engagement in children's education, particularly for families with no formal education. However, the strategies and interventions to optimize caregiver engagement and child outcomes must be culturally and contextually driven [90] and tailored to match caregivers' educational backgrounds, capacities, and circumstances if they are to be effective. Leveraging longitudinal and qualitative methodologies, such co-created, context-specific, tailored approaches can contribute to reducing the mental health burden of Ghanaian parents and increasing their children's developmental potential.

## Acknowledgments

We thank the partners who helped develop and implement the program, including Movva technologies, NORSSAC, Plan International, Esinam Avornyo, as well as Edward Tsinigo and Richard Murphy Edro from Innovations for Poverty Action Ghana for support for data collection, as well as the team of data collectors who collected data in spite of different challenges. We are also incredibly grateful to the children and caregivers that participated in the intervention.

## Author Contributions

**Conceptualization:** Marilyn N. Ahun, Richard Appiah, Elisabetta Aurino, Sharon Wolf.

**Data curation:** Sharon Wolf.

**Formal analysis:** Marilyn N. Ahun.

**Funding acquisition:** Marilyn N. Ahun.

**Investigation:** Marilyn N. Ahun, Richard Appiah, Elisabetta Aurino, Sharon Wolf.

**Writing – original draft:** Marilyn N. Ahun.

**Writing – review & editing:** Marilyn N. Ahun, Richard Appiah, Elisabetta Aurino, Sharon Wolf.

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
