## [Decision Letter · Decision Letter 0]

4 Jul 2024

PGPH-D-24-00810

Caregiver mental health and school-aged children’s academic and socioemotional outcomes: Examining associations and mediators in Northern Ghana

Dear Dr. Ahun,

Thank you for submitting your manuscript to PLOS Global Public Health. After careful consideration, we feel that it has merit but does not fully meet PLOS Global Public Health’s publication criteria as it currently stands. Therefore, we invite you to submit a revised version of the manuscript that addresses the points raised during the review process.

Please carefully address all of the reviewers comments and revision suggestions in your revised submission, particularly regarding the methods, study limitations and the potential implications of these conclusions.

We look forward to receiving your revised manuscript.

Kind regards,

Jennifer Tucker, PhD

Staff Editor

Journal Requirements:

Additional Editor Comments (if provided):

1. We ask that a manuscript source file is provided at Revision. Please upload your manuscript file as a .doc, .docx, .rtf or .tex.

Reviewers' comments:

Reviewer's Responses to Questions

**Comments to the Author**

1. Does this manuscript meet PLOS Global Public Health’s publication criteria? Is the manuscript technically sound, and do the data support the conclusions? The manuscript must describe methodologically and ethically rigorous research with conclusions that are appropriately drawn based on the data presented.

Reviewer #1: Yes

Reviewer #2: Yes

Reviewer #3: Yes

2. Has the statistical analysis been performed appropriately and rigorously?

Reviewer #1: Yes

Reviewer #2: No

Reviewer #3: Yes

3. Have the authors made all data underlying the findings in their manuscript fully available (please refer to the Data Availability Statement at the start of the manuscript PDF file)?

Reviewer #1: No

Reviewer #2: Yes

Reviewer #3: Yes

4. Is the manuscript presented in an intelligible fashion and written in standard English?

Reviewer #1: Yes

Reviewer #2: Yes

Reviewer #3: Yes

5. Review Comments to the Author

Reviewer #1: In their paper, “Caregiver mental health and school-aged children’s academic and socioemotional outcomes: Examining associations and mediators in Northern Ghana”, the authors utilized a sample of 4714 children in three rural regions of northern Ghana and employed descriptive statistics, linear regressions, and SEM analyses. The primary findings are that the mechanisms through which caregiver mental health is associated with children’s developmental outcomes in Ghana, as well as how mechanisms differ as a function of caregivers’ educational attainment. I believe that this study is worthy of publication in the PLOS Global Public Health with the following considerations.

General

The paper is well written. But, I did see several typos. Therefore, I suggest that the authors engaged a professional editor to make sure the manuscript is as error-free as possible.

Introduction and literature review

Expand the literature review to include more international studies. This can help contextualize the findings within a global framework and highlight the relevance of the study across different cultural settings.

Methods

While the study uses established scales, more information on their validity and reliability in the specific context is needed.

The approaches to dealing with missing data should be reported.

Discussion

The discussion does not sufficiently cover the practical applications of the findings. There need to be more discussions in this section. To as great as extent possible, make sure all of the explanations are also clarifying or expanding on the literature cited in the Introduction and Literature Review section.

In summary, then, I am supportive of this manuscript moving forward towards publication. I recommend “accept with major revisions”. I would be more than happy to review any revised manuscript, if the editor needs may help.

Reviewer #2: 1、 It is recommended to add a literature review in the introduction section. That is, what research currently exists on the relationship between caregiver depression and child development. This will help readers understand the current research status.

2、 Although this study is a randomized intervention experiment, the sampling still has a self-selection bias. If this study is a randomized intervention experiment, it is necessary to show the balance test of the basic data. Check whether the baseline personal characteristics and outcome variables are balanced. And inform us if the intervention group and the control group were selected through random assignment.

3、 It is recommended that the author state the reliability and validity of all the scales used. This will allow readers to understand the effectiveness of these scales.

4、 This study mainly analyzes the relationship between caregiver mental health and child development. There is a strong endogeneity issue, such as the bidirectional causality between caregiver mental health and child development. This study mainly analyzes the direct and indirect effects between the two through structural models. Therefore, it is recommended that the author conduct an endogeneity analysis.

5、 It is recommended that the author test whether there are significant differences in child outcome variables and personal characteristics between caregivers who received education and those who did not in Table 1. It is also suggested that the author first test the average effect and then conduct separate analyses for the groups where caregivers received education and those who did not.

Reviewer #3: I commend the authors for a paper that outlines the mediating role of caregiver engagement in the association between caregiver psychological distress and children’s outcomes. This study is important given that there is limited research on the underlying mechanisms of these associations in LMICs, and in particular Ghana, which is the study’s country of focus. Their results highlight the importance of using culturally sensitive approaches in early intervention programs.

My comments are as follows:

Keywords: The authors should consider using key words that make their paper more likely to be picked up through a search. The words they have used are very generic.

Introduction

Line 64: Please indicate the source from which the statistics for the human capital index are obtained

Line 71: It may be worthwhile to provide comparative information on global estimates in relation to mental health problems among children in Ghana

Line 73: The authors should attempt to link the idea in this paragraph to the previous one

Line 81: It is not clear what ‘it’ refers to in the statement that begins with ‘It is a key input...’

Line 87: The phrase ‘negatively influence’ appears twice in the same statement which makes it cumbersome to read. The statement should be shortened to improve clarity

Methods

The subsection on study population has information on study site, intervention, study design etc. The authors could consider having other subsections under this to help organize the section and make it more navigable for the reader

Line 145: The ethics statement should be written chronologically in the order of obtaining ethics approval – first from the ethics body and then from the study participants. Was there any permission required at the national and subnational levels?

Line 160: I suggest that ‘We therefore...’ be the beginning of a new sentence

What was the rationale for using one SD as the cut-off?

Line 165: What does this statement mean? ‘Children’s academic skills were assessed using assessments adapted by existing scales.’

Line 170: IDELA does not appear on the list of abbreviations. Kindly check that all abbreviations are included on the list

Line 184: The authors should indicate if the International Social-Emotional Learning Assessment has been validated in Ghana or other LMICs

Line 239: What fit indices were considered when performing the SEM?

Results

Line 256: The authors need to provide information on the ethnic groups in Ghana so that there is some context for the information provided on ethnicity. Was ethnic group one an important consideration for the analysis?

Line 264: ‘The prevalence of elevated caregiver psychological distress was 14%, with a slightly higher – but not significantly different (2 = 1.99, p=0.157) – prevalence in caregivers who had no formal education (15%) compared to caregivers that had some formal education (13%).’ This could be stated as ‘There was no significant difference in the prevalence of elevated caregiver psychological distress among those who had no formal education (15%) and those who had some formal education (13%)’ or something along those lines

Line 281: ‘were’ should be ‘where’ The entire paper should be checked for typographical and grammatical errors to improve quality

Line 291: Were the children exposed to higher levels of caregiver psychological distress or is it that they lived in households where their caregivers had higher levels of psychological distress?

Like I indicated earlier, the model fit indices for the SEM should be reported within the text so that the reader can make a judgement on the outcome of the analysis

Discussion

The discussion section could be improved by consistently providing possible explanations for the results presented in the paper. This has been done for some of the results and not for others.

Line 415: What other dimensions of caregiving could be considered?

6. PLOS authors have the option to publish the peer review history of their article (what does this mean?). If published, this will include your full peer review and any attached files.

**Do you want your identity to be public for this peer review?** For information about this choice, including consent withdrawal, please see our Privacy Policy.

Reviewer #1: No

Reviewer #2: No

Reviewer #3: **Yes: **Patricia Kitsao-Wekulo

---

## [Decision Letter · Decision Letter 1]

27 Aug 2024

Caregiver mental health and school-aged children’s academic and socioemotional outcomes: Examining associations and mediators in Northern Ghana

PGPH-D-24-00810R1

Dear Dr. Ahun,

We are pleased to inform you that your manuscript 'Caregiver mental health and school-aged children’s academic and socioemotional outcomes: Examining associations and mediators in Northern Ghana' has been provisionally accepted for publication in PLOS Global Public Health.

Best regards,

Catherine Elizabeth Draper

Academic Editor

Reviewer Comments (if any, and for reference):

Reviewer's Responses to Questions

**Comments to the Author**

1. If the authors have adequately addressed your comments raised in a previous round of review and you feel that this manuscript is now acceptable for publication, you may indicate that here to bypass the “Comments to the Author” section, enter your conflict of interest statement in the “Confidential to Editor” section, and submit your "Accept" recommendation.

Reviewer #1: All comments have been addressed

Reviewer #2: All comments have been addressed

Reviewer #3: All comments have been addressed

2. Does this manuscript meet PLOS Global Public Health’s publication criteria? Is the manuscript technically sound, and do the data support the conclusions? The manuscript must describe methodologically and ethically rigorous research with conclusions that are appropriately drawn based on the data presented.

Reviewer #1: Yes

Reviewer #2: Yes

Reviewer #3: Yes

3. Has the statistical analysis been performed appropriately and rigorously?

Reviewer #1: Yes

Reviewer #2: Yes

Reviewer #3: Yes

4. Have the authors made all data underlying the findings in their manuscript fully available (please refer to the Data Availability Statement at the start of the manuscript PDF file)?

Reviewer #1: Yes

Reviewer #2: Yes

Reviewer #3: No

5. Is the manuscript presented in an intelligible fashion and written in standard English?

Reviewer #1: Yes

Reviewer #2: Yes

Reviewer #3: Yes

6. Review Comments to the Author

Reviewer #1: (No Response)

Reviewer #2: no other comments

Reviewer #3: The authors have addressed the comments that I raised adequately through the revisions they have made to the manuscript

7. PLOS authors have the option to publish the peer review history of their article (what does this mean?). If published, this will include your full peer review and any attached files.

**Do you want your identity to be public for this peer review?** For information about this choice, including consent withdrawal, please see our Privacy Policy.

Reviewer #1: No

Reviewer #2: No

Reviewer #3: **Yes: **Patricia Kitsao-Wekulo
